

# Snow heterogeneous reactivity of bromide with ozone lost during snow metamorphism

Jacinta Edebeli[1,2], Jürg C. Trachsel[3], Sven E. Avak[1], Markus Ammann[1], Martin Schneebeli[3], Anja Eichler[1,4], Thorsten Bartels-Rausch[1]

[1]Laboratory of Environmental Chemistry, Paul Scherrer Institut, Villigen PSI, Switzerland
[2]Swiss Federal Institute of Technology, ETH Zurich, Zürich, Switzerland
[3]WSL-Institute for Snow and Avalanche Research SLF, Davos Dorf, Switzerland
[4]Oeschger Centre for Climate Change Research, University of Bern, Bern, Switzerland

*Correspondence to*: Thorsten Bartels-Rausch (thorsten.bartels-rausch@psi.ch)

**Abstract.** Earth's snow cover is very dynamic on diurnal time scales. The changes to the snow structure during this metamorphism have wide ranging impacts such as on avalanche formation and on the capacity of surface snow to exchange trace gases with the atmosphere. Here, we investigate the influence of dry metamorphism, which involves fluxes of water vapor, on the chemical reactivity of bromide in the snow. For this, the heterogeneous reactive loss of ozone at a concentration of 5-6 × $10^{12}$ molecules cm$^{-3}$ is investigated in artificial, shock-frozen snow samples doped with 6.2 μM sodium bromide and with varying metamorphism history. The oxidation of bromide in snow is one reaction initiating polar bromine releases and ozone depletions. We find that the heterogeneous reactivity of bromide is completely absent from the air-ice interface in snow after 12 days of temperature gradient metamorphism and suggest that burial of non-volatile bromide salts occurs when the snow matrix is restructuring during metamorphism. Impacts on polar atmospheric chemistry are discussed.

## 1 Introduction

Snow on Earth hosts chemical reactions that impact the composition of the atmosphere (Dominé and Shepson, 2002; Grannas et al., 2013). One example is the oxidation of bromide and the subsequent release of bromine from arctic snow (Abbatt et al., 2010; Saiz-Lopez and von Glasow, 2012). This reactive halogen species participates in ozone destroying chemical cycles in the gas phase. Ozone is one of the main oxidants in the lower atmosphere with impact on atmospheric composition, health, and climate (Simpson et al., 2007). Recent improvement in global atmospheric chemistry models indicate that halogen chemistry is responsible for about 14% of the global tropospheric $O_3$ reduction (Schmidt et al., 2016). In addition, the reactive halogen species are potent oxidants for organics and, of particular interest, gas phase mercury (Simpson et al., 2007; Simpson et al., 2015). Oxidized mercury partitions readily into condensed phases from where it may enter the ocean and the food-web upon seasonal snow melt (Steffen et al., 2008).



Dominé et al. (2008) argued that the efficient chemical reactivity in snow is linked to its physical properties. Snow is a porous
matrix that is dense enough to provide a large surface area for heterogeneous reactions, but not too dense to limit transport and
light penetration as seen in soil, for example. The heterogeneous oxidation of bromide by ozone, a potential pathway for
bromine release both in the dark and in sunlight (Abbatt et al., 2010), has been shown to be very efficient on ice and brine
surfaces (Wren et al., 2010; Oldridge and Abbatt, 2011; Edebeli et al., 2019). The high rates have been linked to an ozonide
intermediate and its stabilisation at the surface (Artiglia et al., 2017). Consequently, the location of chemical reactants - their
distribution between the air-ice interface and other reservoirs in the interior of the snow - is a key determinant for their chemical
reactivity (Bartels-Rausch et al., 2014; Hullar and Anastasio, 2016; McFall et al., 2018). Field studies have revealed a high
heterogeneity in bromine release and bromide concentration in snow and have attributed this heterogeneity to the initial source
of bromide and to post-depositional changes of the location (Jacobi et al., 2012; Pratt et al., 2013).
One prominent post-depositional mechanism is dry metamorphism shaping the structure and physical properties of snow with
impact on heat transfer, albedo, and avalanche formation (Blackford, 2007; Dominé et al., 2008; Schweizer, 2014). Snow at
Earth's surface that is exposed to varying temperature gradients with time undergoes continued sublimation and deposition
during metamorphism with complete re-building of the entire snow matrix every few days (Pinzer and Schneebeli, 2009a;
Pinzer and Schneebeli, 2009b). Earth's snow cover can be exposed to temperature gradients between $10\ \mathrm{K\ m^{-1}}$ to $100\ \mathrm{K\ m^{-1}}$
(Birkeland et al., 1998). Dominé et al. (2015) showed that such temperature gradient conditions can prevail on a seasonal scale:
in low-arctic tundra, snow is exposed to a temperature gradient mostly above $20\ \mathrm{K\ m^{-1}}$ between mid-November and early
February. The consequences are changes in the isotopic composition of the snow with implications for ice core dating (Steen-
Larsen et al., 2013; Steen-Larsen et al., 2014). Further, Hagenmuller et al. (2019) observed dust particles being incorporated
into the ice matrix of snow driven by the intensive water vapor fluxes during dry, temperature gradient metamorphism. With
the turnover of snow grains and the movement of water vapor, contaminants may be redistributed between the surface and
bulk of the snow grains: Studies investigating the adsorption and uptake of trace gases such as nitric acid and hydrochloric
acid with growing ice have observed higher uptake than in ice at equilibrium (Kärcher and Basko, 2004; Ullerstam and Abbatt,
2005; Kippenberger et al., 2019). Kippenberger et al. (2019) has shown that the burial of volatile acids is a strong function of
acidity, growth rate, and temperature. At equilibrium, adsorption of acidic trace gases leads to the acids or their anions entering
the ice phase at considerable concentration only within the interfacial region of a few nm depth, as recently observed for
hydrochloric acid and volatile organic acids (Krepelova et al., 2013; Bartels-Rausch et al., 2017; Kong et al., 2017; Waldner
et al., 2018). Therefore, recrystallization in snow might have a significant impact on the fraction of contaminants or reactants
located at the air-ice interface of snow and thus on the heterogeneous chemistry of ions in snow. Laboratory studies
investigating temperature gradient metamorphism effects in natural and artificial snow have observed a strong influence of
metamorphism on the elution behaviour of ions such as ammonium, fluoride, chloride, calcium and sulphate. Whereas calcium



and sulphate were found to be enriched at the air-ice or ice-ice interface during snow metamorphism, ammonium, fluoride,
and chloride were buried in the bulk of the snow (Hewitt et al., 1989, 1991; Cragin et al., 1996; Trachsel et al., 2019).
Here, we study the effect of sublimation and growth of ice during snow metamorphism on bromide reactivity in well controlled
laboratory experiments. The sodium bromide used in this study is non-volatile and field studies have related its mobility in the
snowpack to its vivid photochemical transformation into volatile bromine. Bromine is released to the air and may re-deposit
on the snow surface after formation of stickier species, such as HOBr (Toom-Sauntry and Barrie, 2002).

The objective of this study is to investigate the heterogeneous reactivity of bromide oxidation by gas-phase ozone to assess the
surface concentration of bromide and its change during temperature gradient metamorphism. Bromide concentration in the
doped snow samples (6.2 µM) is on the lower end of observations in environmental snow (Krnavek et al., 2012), but slightly
higher than that observed in snow in the Arctic (Dibb et al., 2010).
**Experimental**
Snow samples were prepared by shock-freezing aqueous solutions (Bartels-Rausch et al., 2004; Trachsel et al., 2019) and
stored in a metamorphism box with a well-defined temperature gradient at the WSL Institute for Snow and Avalanche Research
SLF in Davos (Trachsel et al., 2019). After the exposure to the temperature gradient, the individual samples were exposed to
ozone in a packed-bed flow tube set-up to derive the impact on the reactivity with gas-phase ozone (Bartels-Rausch et al.,
2004). The structure of snow samples before and after metamorphism was imaged by X-ray microtomography (Trachsel et al.,

79   2019).

**Sample preparation**
Artificial snow was produced by shock freezing droplets of a sample solution in liquid nitrogen. The sample solution was
either ultrapure water (18 MΩ quality, arium pro, Sartorius, Göttingen, Germany) (undoped snow) or 640 ppb sodium bromide
(NaBr, Sigma Aldrich, >99.0%) in ultrapure water (doped snow). The samples were left overnight at –45°C and then, stored
isothermally at –5 °C for 7 days to anneal. The samples were returned to –45°C after this isothermal treatment to slow down
further changes with time. The snow was sieved using pre-cleaned stainless-steel sieves (Retsch, Germany) in a –20°C cold
laboratory at the WSL Swiss Snow and Avalanche research Institute (SLF, Davos, Switzerland). Snow grains in the size range
300 – 600 µm were packed into the $12.0 \pm 0.1$ cm long glass reactor tubes with $2.4 \pm 0.1$ cm internal diameter. All samples
were stored isothermally at -5 °C for 7 days to minimize grain-boundaries and up to 54 days at - 45 °C prior to the
metamorphism experiments for logistic reasons (see Results and Discussion). The bromide concentration in the sieved snow
crystals was $6.2 \pm 0.18$ µM ($498 \pm 14$ ppbw) (doped snow) and <0.12 µM (undoped snow) as determined by ion chromatography
(Metrohm (Herisau, Switzerland) 850 Professional IC, 872 Extension Module, 858 Professional Sample Processor
autosampler). A Metrosep A Supp 10 column (Metrohm) was used and the eluents were a 1.5 mM $Na_2CO_3$ and 0.3 mM



NaHCO$_3$ in a 1:1 mixture followed by 8 mM Na$_2$CO$_3$ and 1.7 mM NaHCO$_3$ in a 1:1 mixture with a flow rate of 0.9 ml min$^{-1}$.
Possible instrumental drifts were monitored by measuring a standard after every 20th sample.

**Metamorphism**

For the temperature gradient metamorphism experiments, samples were exposed to a gradient of 31 K m$^{-1}$ for 12 days in a
snow metamorphism box mounted in a cold room at –8 °C (at SLF, Davos, Switzerland). The metamorphism box was a heavily
insulated box with a heating plate set to –4 °C at the bottom. Over this plate, there was a ~ 2-3 cm thick layer of ice from
ultrapure water. The sample holders were mounted on a disk with a 0.5 cm layer of ice made with ultrapure water in contact
with the snow grains to increase thermal contact (Pinzer and Schneebeli, 2009a). The spaces between the sample tubes were
filled by sieving in snow. The box was then covered with a thin plastic film in contact with the filled-in snow and caps of the
samples to avoid losses due to sublimation. This set-up resulted in an effective temperature at the bottom and at the top of the
snow samples of -4.4 ± 0.1 °C and -8.1 ± 0.1 °C. After the temperature gradient metamorphism treatment, the samples were
stored at –45°C. For comparison, additional samples were stored isothermally at -20 °C at SLF, Davos, Switzerland for 12
days. In total, 12 samples were prepared from the homogenized snow batches: 2 undoped and 2 doped samples that experienced
12-days temperature gradient metamorphism, 2 undoped and 2 doped samples without temperature gradient metamorphism, 2
undoped and 2 doped samples that experienced iso-thermal metamorphism. The replica of the doped snow that was exposed
to temperature metamorphism for 12 days and of the undoped snow that was not exposed to temperature gradient
metamorphism could not be analysed due to technical failures during the experiments.
Structural changes in the samples were assessed using an X-ray computer micro-tomography scanner (Scanco micro-CT 40)
with a resolution of 10 μm. This microCT was operated at –20°C. Details of operations of the microCT scans have been
described by Pinzer and Schneebeli (2009a). The reconstructed microCT images were filtered with a Gaussian filter (support
2 voxels, standard deviation 1 voxel) and the threshold for segmentation was applied according to Hagenmuller et al. (2014).
Structural parameters of the segmented ice structure were extracted with the software tools of the microCT device (Image
Processing Language, Scanco Medical) to calculate the porosity and specific surface area.

**Packed bed flow tube experiments**

Samples were exposed to ozone at –15°C. Before exposure, about 2 cm of the samples were scraped off from the top and
bottom of the samples to avoid potential contamination from contact with the ice layer on the disk in the metamorphism box
or the caps for the sample holder/reactor tubes. An exception to this is one of the 0-day doped samples where 3 cm were shaved
off. Afterwards, the mass of each snow sample during the ozone exposure was determined based on the weight of the filled
and empty sample tube. The sample tubes were placed in the reactor cell, an insulated cooling jacket, at –15°C. The sample
was allowed to temperature equilibrate for an hour before exposure to gases. Humidified airflow of ~200 ml min$^{-1}$ O$_2$ and



~200 ml min$^{-1}$ N$_2$ was delivered through the sample for 30 minutes to condition the sample. The total flow rate through the
sample was set between 339 ml min$^{-1}$ to 352 ml min$^{-1}$ at norm temperature and pressure of 273.15 K and 1013.25 bar. This
airflow was humidified to a water vapor pressure of ice at $-15.0 \pm 0.3$ °C. Ozone was generated by passing the N$_2$/O$_2$ airflow
through a pen ray Hg UV lamp. The ozone flow was also humidified before delivery to the sample. The flow was alternated
between a bypass and the sample to control for drifts in ozone concentration. Ozone concentration was monitored using a
commercial analyser (Teledyne, model 400E). The average ozone concentration for each experiment was slightly different due
to the day to day variability in the efficiency of the ozone generator. For all experiments, ozone concentrations varied from
163 to 212 ppb ($4.7$-$6.2 \times 10^{12}$ molecules cm$^{-3}$). The maximum variability during any one experiment was less than 5 ppb after
attaining initial stability at the start of the experiment. This drift was accounted for during analysis using fitting routines. To
confirm perfect flow conditions in the packed bed flow tubes, the chromatographic retention of acetone was determined for
some samples at -30°C. Once the ozone experiment was finished, the samples were exposed to a flow of acetone in humidified
N$_2$ (Bartels-Rausch et al., 2004). The observed retention time of acetone at -30°C matched calculations based on the air-ice
partitioning coefficient (Dominé and Rey-Hanot, 2002; Winkler et al., 2002; Peybernes et al., 2004; Bartels-Rausch et al.,
2005; Crowley et al., 2010) and the specific surface area of the snow sample as derived by microCT measurements for the
undoped and doped samples after temperature gradient metamorphism.

## 1 Results and Discussion

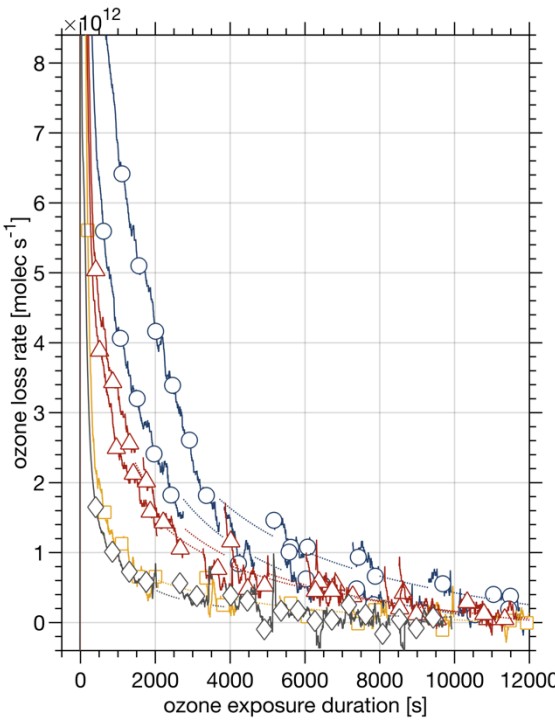

**Figure 1: Ozone loss rate with duration of exposure.** The snow samples with a bromide concentration of 6.2 μM experienced 0 days (blue lines, open circles) and 12 days (yellow line, open squares) of temperature gradient metamorphism with a temperature gradient of 31 K m$^{-1}$. The dotted lines are guide to the eyes, for periods where ozone loss data are not available (see text for details). Also shown are the ozone loss rates of snow samples after 12 days of isothermal metamorphism at -20 °C (red lines, open triangles). The grey line (open diamonds) denotes the average ozone loss rates of 5 undoped samples with and without exposure to temperature gradient metamorphism. The gas phase mixing ratio of ozone varied between 4.7-6.2 × 10$^{12}$ molecules cm$^{-3}$ for individual samples. Temperature during ozone exposure was -15 °C.

Figure 1 shows ozone loss rates for snow samples prior to and after exposure to temperature gradient metamorphism. The ozone loss rate was derived based on observed changes in gas-phase ozone concentration downstream of the flow tube packed with the snow sample. The ozone loss rate is largest for the two samples doped with 6.2 μM bromide prior to ageing under laboratory-controlled dry metamorphism with a constant temperature gradient of 31 K m$^{-1}$ with $4 \times 10^{12}$ molecules s$^{-1}$ and $7 \times 10^{12}$ molecules s$^{-1}$ at 1000 s duration of ozone exposure (Fig. 1, blue lines, open circles). The differences in ozone loss rate of these two samples can be assigned to variations in sample mass and in the amount of bromide at the air-ice interface (see below). The loss rate was reduced by a factor of about 4-7 in the snow sample that experienced temperature gradient metamorphism with $1 \times 10^{12}$ molecules s$^{-1}$ at 1000 s duration of ozone exposure (Fig 1, yellow line, open square). This loss rate is indistinguishable from that in the samples without added bromide with a mean of $1 \times 10^{12}$ molecules s$^{-1}$ at 1000 s for 5 samples and with a standard deviation of $0.4 \times 10^{12}$ molecules s$^{-1}$ at 1000 s (Fig. 1, grey line, open diamonds). This observed



loss is attributed to the reaction of ozone with traces of impurities. Furthermore, the residence time of the ozone gas in the
porous snow structure contributes to the apparent loss rate at the start of the experiments. Also shown is the loss rate from 2
samples that experienced isothermal metamorphism for 12 days at -20 °C (Fig. 1, red lines, open triangles). The loss rate is
reduced compared to the samples before exposure to metamorphism. Taken the large variation in the ozone loss of samples
that were not exposed to metamorphism, we refrain from discussing this difference further. Despite the uncertainty caused by
the variation in observed ozone loss, the ozone loss in samples without exposure to temperature gradient metamorphism (Fig.
1, blue and red lines) are significantly higher than the loss rate after temperature gradient metamorphism. Before we elaborate
on the mechanism of this loss, we start by discussing details of the apparent loss rates.

The loss rate prior to temperature gradient metamorphism ($4\text{-}7 \times 10^{12}$ molecules s$^{-1}$) agrees well with loss rates of 2-
$6 \times 10^{12}$ molecules s$^{-1}$ as derived based on earlier experimental work. Oldridge and Abbatt (2011) reported an uptake
coefficient of $1.5 \times 10^{-8}$ in coated wall flow tube studies on frozen sodium bromide/sodium chloride/water mixtures at -15°C
and Wren et al. (2010) reported $4 \pm 2 \times 10^{-8}$ in a laser-induced fluorescence study with sodium bromide/water mixtures at
- 20°C. The uptake coefficient normalizes the loss rate to the collision rate of ozone with the surfaces. In this work, we refrain
to report the results as uptake coefficient, as only the surface area of the snow is known, but not the surface area covered with
reactive sodium bromide (see below). To compare to our work, the reported uptake coefficients were transferred into loss rates
based on the specific surface area of the snow sample used in this work and an ozone concentration of $4.7\text{-}6.2 \times 10^{12}$ molecules
cm$^{-3}$. The studies by Wren et al. (2010) and by Oldridge and Abbatt (2011) were done with an initial sodium bromide
concentration of 10 mM and a gas-phase ozone concentration of $1 \times 10^{14}$ molecules cm$^{-3}$ and $80 \times 10^{14}$ molecules cm$^{-3}$,
respectively. The concentration of sodium bromide in the reactive solutions in equilibrium with ice is a sole function of
temperature, and thus identical even for our samples that were frozen from aqueous solutions with 6.2 µM bromide.
Uncertainty in this comparison comes from the very low ozone concentration of $5 \times 10^{12}$ molecules cm$^{-3}$ used in this study.
Based on the results by Oldridge and Abbatt (2011), one would expect increasing uptake coefficients with lower ozone
concentrations that can be assigned to a surface reaction. In summary, we conclude that the oxidation of bromide by ozone
leads to the loss of ozone in the initial period of the experiments. Figure 1 further shows how the ozone loss rates strongly
decrease with the duration of ozone exposure. After about 8000 s ozone exposure, the raw data curves levelled off approaching
a loss rate of $1.1\text{-}1.9 \times 10^{12}$ molecules s$^{-1}$. Please note, that this loss rate has been subtracted from the data discussed and shown
in Fig. 1. This background loss rate is attributed to the ozone self-reaction on the ice surface. Support comes from earlier work
by Langenberg and Schurath (1999) describing a reactive ozone uptake coefficient on ice of $7.7\text{-}8.6 \times 10^{-9}$ at -15 °C and at
ozone gas-phase concentrations similar to our work. A loss rate of $0.86\text{-}0.90 \times 10^{12}$ molecules s$^{-1}$ can be derived based on the
reported uptake coefficient for the experimental conditions of our doped samples prior to metamorphism, in perfect agreement
with our observations stated above.





The cumulative loss of ozone is $0.9\text{-}1.7 \times 10^{16}$ molecules for snow doped with 6.2 μM bromide without exposure to
metamorphism and $6.3 \times 10^{14}$ molecules for the doped sample after exposure to 12 days temperature gradient metamorphism.
The cumulative loss was derived by integrating the area below the loss rate curves in Fig. 1 between 500 and 8000 s and
subtracting the cumulative loss of the undoped sample to account for the presence of impurities also in the samples doped with
bromide. For this analysis, the missing data in periods where the carrier gas was bypassing the snow to monitor the ozone
concentration delivered to the flow tube were estimated using a power fit to the data (Figure 1). Now that we have established
the ozone loss rate and the number of ozone molecules lost in total, we address the amount of bromide that is oxidised by the
ozone. Generally, the products and reaction mechanism of the bromide oxidation by ozone in the aqueous phase strongly
depend on reaction time, reactant concentration and pH (Haag and Hoigne, 1983; Heeb et al., 2014). For non-acidified
conditions, as in our study, hypobromous acid ($HOBr/OBr^-$) is the main product (Eq. 1) that may react further with ozone (Eq.
2) to form bromite ($BrO_2^-$), disproportionate to bromide ($Br^-$) and bromate ($BrO_3^-$), or self-react to dibromine monoxide ($Br_2O$)
(Heeb et al., 2014). Despite uncertainties in the precise product distribution in this study, ozone is lost in our study in the initial
reaction with bromide and to some extent in the subsequent oxidation of hypobromous acid to bromite resulting in 1-2 ozone
molecules lost per bromide molecule. In particular at acidic conditions as relevant for atmospheric waters and ices (Abbatt et
al., 2012; Bartels-Rausch et al., 2014), bromine is formed and released to the atmosphere in a sequence of reaction steps (Eqs.
1 and 2) that consume 0.5 ozone molecules per bromine molecule (Abbatt et al., 2012). The release of bromine has also been
observed in experiments with frozen sea-salt mixtures that contain bromide (Sjostedt and Abbatt, 2008; Oldridge and Abbatt,

209   2011).


$Br^- + O_3 \longrightarrow OBr^- + O_2$ (Eq. 1)
$OBr^- + O_3 \longrightarrow BrO_2^- + O_2$ (Eq. 2)
$OBr^- + Br^- + H^+ \longrightarrow Br_2 + OH^-$ (Eq. 3)

Thus, assuming a net loss of 1 ozone molecule per bromide molecule, one might estimate about $0.9\text{-}1.7 \times 10^{16}$ molecules of
bromide are available for the multiphase reaction with ozone in the porous snow prior to metamorphism. To put this number
into perspective, this amount of bromide corresponds to a formal surface concentration of $4\text{-}5 \times 10^{12}$ molecules cm$^{-2}$ assuming,
for comparison reason, that the bromide is located at the surface. Taken that the adsorption of most trace gases can be described
by a Langmuir isotherm saturating at around $3 \times 10^{14}$ molecules cm$^{-2}$ (Abbatt, 2003), the formal Langmuir surface coverage
would be approximately 1 %. This low coverage supports the argument that the decreasing trend of the ozone loss rates with
duration of ozone exposure observed for the doped samples prior to metamorphism is caused by depletion of the available
bromide through the oxidation by ozone. The cumulated amount of reacted bromide can further be compared to the total
amount of bromide of $4\text{-}6 \times 10^{16}$ molecules initially added to the snow sample. Apparently, 22 % - 26 % of the bromide was
accessible to gas-phase ozone, the majority of bromide was not available for reaction prior to metamorphism.






This result raises the question of the initial location and phase of the sodium bromide in the shock-frozen, artificial snow
samples. Shock freezing aqueous solutions may preserve the homogeneous distribution of solutes also in the grains. With the
low aqueous concentration of 6.2 μM and a diffusivity of solutes in ice of $100 \times 10^{-12}$ cm$^2$ s$^{-1}$, one may estimate that the total
amount of bromide diffusing from the ice to the surface where it reacts with ozone is $1.6 \times 10^{10}$ molecules each second. This
is much less than the ozone loss observed in our experiments clearly showing that the bromide is not present in the snow
samples as homogeneous solid-solution. Due to lack of diffusion rates of bromide in ice, the diffusion rates of HNO$_3$ in
crystalline ice at -15 °C of $100 \times 10^{-12}$ cm$^2$ s$^{-1}$ (Thibert and Dominé, 1998) was used as upper limit in this calculation.
Interestingly, the data by Dominé and co-workers also allow to estimate solubility of sodium bromide in ice as solid solution,
that is in thermodynamic equilibrium. In their well-controlled experiments, Thibert and Dominé (1997, 1998) derived
solubilities of up to 0.1 mM to 1 mM for HCl at 265 K to 238 K and up to 0.06 mM to 0.6 mM for HNO$_3$ in ice. These data
describe the equilibrium between gas-phase acid and solid solution and may serve as estimate for the solubility limit of sodium
bromide in ice. Clearly, the apparent concentrations of 6.2 μM used in the experiments described here is lower than the
estimated solubilities in ice. That we find a significant fraction of bromide at the air-ice interface confirms that freezing
seldomly results in thermodynamic equilibria. The initial distribution of impurities in frozen ice is rather a function of the rate
at which the freezing front proceeds (Cappa et al., 2008; Bartels-Rausch et al., 2014). Exclusions of bromide to the interface
of ice during freezing has been observed by others at higher concentration (Wren et al., 2010). Another reservoir, besides the
air-ice interface, to which solutes in shock-frozen salt solutions are expelled are micropockets. Micropockets have been
observed in natural ice cores, interestingly in the interior of the ice matrix rather than at the ice-ice grain boundaries (Eichler
et al., 2017; Eichler et al., 2019). Detection in shock-frozen solutions in the laboratory is hampered by the sensitivity limit to
detect these features with a diameter of ~2 μm or less in laboratory ice (Hullar and Anastasio, 2016). Hullar and Anastasio
(2016) and McFall et al. (2018) have concluded that in shock-frozen caesium chloride (sodium nitrate) solution with a
concentration of 1mM (50 μM), the brine might accumulate to some extent in micropockets, based on indirect evidence.
Similarly, Wren and Donaldson (2011) have shown, that the brine of a 100 mM magnesium nitrate solution is not completely
expelled to the air-ice interface and suggest that micropockets are present as well. Thermodynamics dictate that the sodium
bromide in the heterogeneous, multi-phase mixtures forms liquid brine with a concentration of 3.4 M (1.6 M) during the ozone
exposure at -15°C (metamorphism with a mean temperature of -6°C). For this calculation, the freezing point depression data
by Stephen and Stephen (1963) and Rumble (2019) was used. The eutectic temperature of sodium bromide is at or below -
28 °C (Stephen and Stephen, 1963). With a total amount of $4\text{-}6 \times 10^{16}$ bromide molecules in the samples, $2\text{-}3 \times 10^{-8}$ l ($4\text{-}7 \times$
$10^{-8}$ l) solution are formed at -15 °C (- 6°C). Interestingly, this total amount of brine would fit into 500 (230) micropockets 1
μm in diameter at -6 °C (-15 °C). Based on this estimate, we cannot exclude the presence of micropockets during
metamorphism and during the ozone exposure in flow tubes in the interior of the ice or at the surface of the ice where they are


often called patches. On the contrary, a homogenous film covering the total snow surface is rather unlikely. Such a brine layer
would have a thickness of only 0.2 nm at -6°C (0.1 nm at -15°C) with a concentration of 3.4 M.

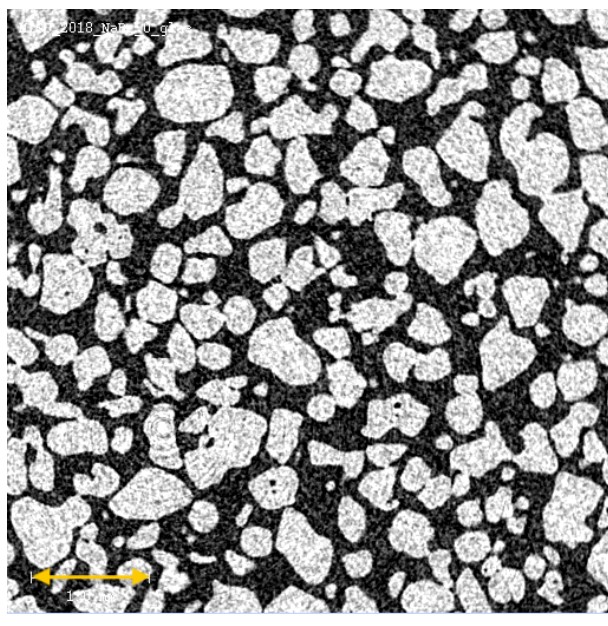


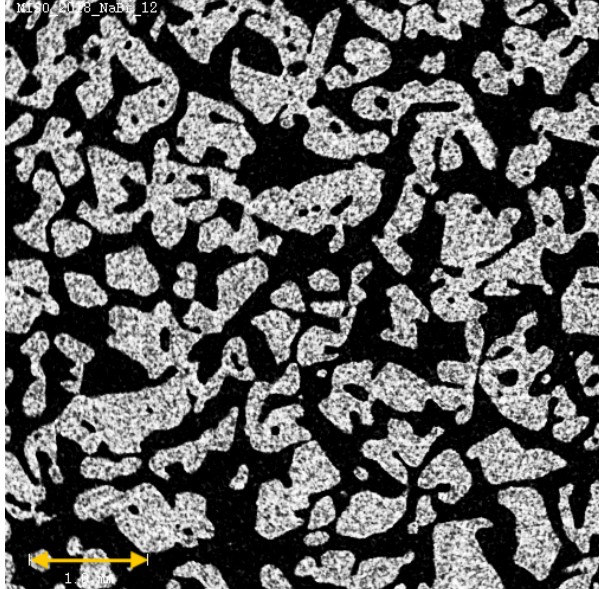

**Figure 2: MicroCT images showing cross-sections of the doped snow samples after 0 days (upper) and 12 days (lower) exposure to**
**temperature gradient metamorphism.** White areas show the ice phase, black represents interstitial air. The scale bar (yellow arrow)
denotes 1 mm.





Despite the uncertainty in the precise initial location of bromide, this study clearly shows that temperature gradient
metamorphism leads to a loss of heterogeneous reactivity with time. We interpret the entire loss of bromide that was initially
available for heterogeneous chemistry to bromide burial driven by the locally growing ice during temperature gradient
metamorphism. The structural changes to the snow during the 12 days temperature gradient metamorphism are visualised by
X-ray microtomography (microCT) images in Fig. 2. During snow metamorphism a coarse and fully connected porous snow
structure grows out of the individual snow particles. This reconstruction is a direct consequence of the temperature gradient in
snow resulting in water vapour pressure gradients which induce fluxes of water vapour from warmer to colder regions. This
gas-phase movement of water is limited to short distances (Yosida et al., 1955). In the experiments described here, the locally
and continuously sublimating and growing snow leads to about 5 complete renewal cycles of the snow structure during the 12-
days temperature gradient metamorphism (Pinzer et al., 2012). Despite the large local water turnover rate, Table 1 shows that
the specific surface area (SSA) did not significantly change during the temperature gradient metamorphism. A convenient side
effect of these little changes is that the kinetic experiments (Figure 1) were done with samples of similar specific surface area.
That changes in SSA do not necessarily reflect water turn-over rates during metamorphism has been discussed before (Pinzer
et al., 2012). The SSA and porosity are within the range observed for hard wind-packed snow and depth hoar in the field
(Legagneux et al., 2002; Zermatten et al., 2011; Calonne et al., 2012). In the microCT image of the snow sample prior to
metamorphism individual spheres with 300 – 600 µm diameter are visible. The particles show edged structures even in absence
of temperature gradient metamorphism (Fig. 2 upper graph).  Samples were stored isothermally at -5 °C for 7 days and up to
54 days at - 45 °C prior to the metamorphism experiments. The tendency to eliminate differences in surface energy is the
driving force in isothermal metamorphism (Dominé et al., 2008; Kerbrat et al., 2008; Löwe et al., 2017); this leads to much
smaller fluxes of water vapour and consequently significantly slower re-structuring compared to temperature gradient
metamorphism (Kämpfer et al., 2005). Consequently, we would not have expected edge growing in the structure. We attribute
this structural change to small but unintended gradients during isothermal storage of the sample. The intention of the isothermal
storage at -5 °C was to allow time to eliminate internal grain boundaries (Blackford, 2007). In line with the lower water vapour
fluxes in isothermal metamorphism, Figure 1 clearly shows that the ozone loss rate is significantly higher in isothermally stored,
doped sampled than that of the undoped samples after 12 days of isothermal metamorphism at -20 °C. Due to the fluctuation
in the ozone loss rate observed in the samples prior to temperature gradient exposure, we refrain from discussing whether the
loss rate after iso-thermal metamorphism at -20 °C is significantly reduced compared to the loss rate observed in samples prior
to metamorphism or if the apparent reduction in loss rate is due to different amounts of bromide available at the surface in the
individual samples.

**Table 1: Morphology of the snow samples; temperature gradient metamorphism age is number of days in the metamorphism box.**
**SSA is specific surface area (± 6% error (Kerbrat et al., 2008). ε is porosity.**





| temperature gradient metamorphism [days] | Bromide [ppbw] | SSA [cm²/g] | ε [-] |
|---|---|---|---|
| 0 | <10 | 176 ± 11 | 0.45 ± 0.005 |
| 12 | <10 | 167± 10 | 0.56 ± 0.01 |
| 0 | 498± 14 | 183± 11 | 0.47 ± 0.01 |
| 12 | 498± 14 | 162 ± 10 | 0.47 ± 0.001 |

The observed burial of bromide during the temperature gradient metamorphism may be attributed to a combination of growing ice, covering the bromide present at the air-ice interface with neat ice, and diffusion of the bromide into the growing ice as described in our previous work (Trachsel et al., 2019). Diffusion rates of bromide in crystalline ice are not known. Diffusion rates of HCl, HNO$_3$, and formaldehyde in crystalline ice at -6 °C range from $7\text{-}240 \times 10\text{-}12$ cm$^2$ s$^{-1}$ (Thibert and Dominé, 1997, 1998; Barret et al., 2011), which allows us to calculate a mean diffusive distance of $40 - 220$ nm s$^{-1}$. This diffusive distance is thus larger than the ice growth rate of 2 nm s$^{-1}$ (Trachsel et al., 2019) supporting the ice-growth diffusion mechanism. A recent study by Wu et al. (2017) showed that bromide is likely to be incorporated in the ice with recrystallization especially at low concentration. Molecular dynamics simulations by Wu et al. (2017) showed that the charge density around a bromide ion does not result in very large disruptions of the local ice structure as observed for other ions such as fluoride. Therefore, they concluded that incorporating bromide into the ice structure may be energetically feasible. Revisiting the micropockets and patches addressed above, one could propose that these micropockets could also be covered by the growing ice in line with Nagashima et al. (2018), who observed preferential growth of ice onto of brine droplets compared to the neat ice surface. The results presented here show that after 5 complete recrystallisation cycles the bromide is absent from the air-ice interface. This depletion of bromide at the air-ice interface is in excellent agreement with previous observations of other ions in snow during metamorphism (Hewitt et al., 1991; Cragin et al., 1996; Trachsel et al., 2019). Elution profiles of shock-frozen snow doped with a mixture of ammonium, calcium, chloride, fluoride, sodium, and sulphate revealed decreasing amounts of all ions at the air-ice interface with duration of snow metamorphism up to 12 days (Trachsel et al., 2019). On longer time scales, calcium and sulphate showed increasing occurrence at the air-ice interface. A further finding from Trachsel (2019) is that the cation and anion tend to experience the same fate in shock-frozen snow. One might thus speculate, that the sodium in the experiments presented here is likewise depleted at the air-ice interface during metamorphism. Cragin et al. (1996) and Hewitt et al. (1991) have shown preferential elution of sulfate compared to chloride and nitrate in snow samples after metamorphism. They proposed that latter ions were incorporated into the ice matrix of snow during dry metamorphism, a finding that was also observed for ammonium and fluoride (Trachsel et al., 2019). A more detailed and quantitative comparison is hampered, as the elution studies generally lack a budget of ions and give no direct link to chemical reactivity. Further, meltwater or the eluent,



induce changes to the snow structure (wet metamorphism) and might lead to relocation of impurities (Meyer and Wania, 2008; Grannas et al., 2013).

**1 Conclusion and Atmospheric Implication**

We have presented an assessment of the effects of metamorphism on the reactivity of ozone with bromide in snow doped with 6.2 μM sodium bromide. Our observation of the ozone consumption showed that the bromide-doped snow samples lost their chemical reactivity towards gas-phase ozone during 12-days of temperature gradient metamorphism. Burial of acidic trace gases with atmospheric relevance has previously been discussed for these volatile species (Huthwelker et al., 2006). Kippenberger et al. (2019) has studied the uptake of HCl and of oxidised organic trace gases to growing ice in Knudsen cell experiments. They observed a continuous uptake only of HCl that exceeded the equilibrium partitioning of HCl to ice (Zimmermann et al., 2016) scaling with ice growth rate and temperature. Growth rates were varied between 2 nm s$^{-1}$ and 110 nm s$^{-1}$. Post-depositional changes to bromide in snow have been observed in the field and have been explained by vivid photochemical reaction into volatile bromine. Volatile bromine might then be re-deposited on the snow surface after formation of more oxidized species, such as HOBr (Jacobi et al., 2002; Toom-Sauntry and Barrie, 2002). In this study, we uniquely show that non-volatile bromide ions are effectively buried. Apparently, temperature gradient metamorphism appears to facilitate the formation of energetically most favourable impurity distributions in snow.

Our findings directly imply that for the Earth surface snow, where temperature gradients are omnipresent, burial of non-volatile solutes during metamorphism can reduce their availability for heterogeneous reactions. That only a small fraction of impurities may be chemically active in surface snow has been discussed for nitrate by Thomas et al. (2011) and Wren and Donaldson (2011). Results from this study thus emphasize that the reactivity of impurities changes dramatically with time during temperature gradient metamorphism in the field, rather than being a result of the initial deposition process. Changes in chemical reactivity with gas-phase species may also hold for those species that were found accumulate at interfaces such as sulphate (Trachsel et al., 2019). Clearly, the tendency to be incorporated into the ice matrix is a strong function of the chemical properties and of concentration (Bartels-Rausch et al., 2014; Trachsel et al., 2019). As a consequence, chemical species that were initially deposited together to the snow might separate to different compartments during metamorphism. The fact that bromide, for example, is driven into the ice while other potential reaction partners might leave the ice may lead to switching off other reaction pathways, such as the oxidation by OH radicals that are produced from organics ending up outside, too far away for the OH to reach the bromide. The driving force for the relocation are temperature inhomogeneities in snow and resulting water vapor fluxes. That ice is not in thermodynamic equilibrium is a frequent situation for atmospheric ice particles as well with common sub- and super- saturation (Gao et al., 2004). Our results therefore suggest that similar re-distribution of ions might also occur prior to snowfall.



In the case of bromide, this re-distribution will suppress an initiation step in bromine explosion and ozone depletion events,
both in light and in the dark, even for snow samples that have an apparently high concentration of bromide. We propose that
this finding -at least partially – explains the varying reactivity of Arctic surface snow. Pratt et al. (2013) has investigated
production of bromine for a range of saline snow and sea ice samples in outdoor chamber experiments and found no correlation
of total bromide concentration in the samples and bromine release. It appeared that pristine snow, where the exchange with the
atmosphere dominates its chemical composition, is more productive than snow that is in contact with sea water. Pratt et al.
(2013) argued that deposition of atmospheric acids to the unbuffered surface snow drives the observed reactivity. Based on
our finding, another explanation would be the constant flux deposition of bromide from the atmosphere refurbishing the buried
bromide and thus providing reactive bromide at the air-ice interface. This finding has significant environmental implications
as it does not only stress the importance of the location of chemical species on their reactivity, but shows that this location is
rapidly changing in surface snow. Further, one should note that incorporation of solutes into the interior of ice and snow makes
them not only resistant to multiphase chemistry, but further reduces their tendency to be washed away by melt- or rain water
percolating the snow. Thus, even under current warming conditions bromide might be a promising candidate for reconstructing
past atmospheric composition from ice core records that have experienced melt effects (Eichler et al., 2001). The enrichment
in the snow may also contribute to later release of toxins to the marine food web upon the complete melting of the snow (Wania
et al., 1998; Eichler et al., 2001; Steffen et al., 2008; Durnford and Dastoor, 2011; Grannas et al., 2013).

## 1 Data availability

Edebeli, Jacinta; Bartels-Rausch, Thorsten (2020). Data set on bromide oxidation by ozone in snow during metamorphism
from laboratory study. EnviDat. *doi:10.16904/envidat.138*.

## 1 Author Contribution

TB-R, AE, MS designed the MISO project that this study was part of. JE planned and performed the flow tube experiments
with help and input from MA, AE, MS, SA, TB-R. JT and JE performed, analysed, and discussed the microCT measurements
with input from MS. TB-R and JE analysed the ozone uptake data and wrote the manuscript with input from MA and all other
authors. All authors approved the submitted version of the manuscript. This work is part of JE doctoral thesis at ETH Zürich.

## 1 Acknowledgement

Funding by the Swiss National Science Foundation (SNSF) under Grant No. 155999 is acknowledged. We thank Matthias
Jaggi (SLF) and Mario Birrer (PSI) for their technical assistance, Margret Matzl (SLF) for her help in evaluating the microCT
data.





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
