# Peer review of "Snow heterogeneous reactivity of bromide with ozone lost during snow metamorphism."

_Atmospheric Chemistry and Physics, 2020_

## Referee Comment (RC1) · Anonymous Referee #3 · 20 May 2020

The manuscript reports on a small series of six experiments quantifying the ozone loss on bromide-doped artificial snow samples. The effect of subjecting the samples to temperature gradients for extended periods of time (days) is studied.

This is an interesting and important study possibly allowing conclusions on the availability of bromide and the processes at the ice-air interface in (aged) snow. It is, therefore, in principle relevant for understanding and modelling bromine release events observed in polar regions.

The manuscript contains important information relevant to the readers of ACP and should be published. However it contains a number of deficiencies and significant improvements are possible and should be made.

[Figure]

1) Frequently release of volatile bromine is mentioned, however the experiments really determine loss of ozone, this fact should be stated more explicitly.

2) The data given in some parts are incomplete and are given in different units, so reading the manuscript requires a pocket calculator. For instance on page 5, lines 125-131 the air flow through the samples is given in ml/minute, the ozone mixing ratio in ppb, while later (page 9) the number of ozone molecules per second is required. Although the manuscript mentions release of bromine 'in light and in the dark' one assumes that the experiments were performed in the dark, but this is not said in the manuscript. Volumes are sometimes given in ml, sometimes in cm3.

3) A table is missing, which summarizes the pertinent data of the experiments: Volume of the reaction chamber, flow rate, snow density, snow surface area, number of ozone molecules lost per second, etc..

4) Fig. 1: The figure summarizes all experimental findings of the manuscript, therefore it should be as informative and clear as possible. However, it is actually quite hard to read since most of the data are huddled in the lowest 20% or so of the plot. It would be helpful if the plot could be split in two, one ranging to 8E12 molec/s or even higher (what are actually the highest measured ozone loss rates?), one showing the data up to e.g. 3E12. Also additional lines indicating the ratio of losses at treated snow vs. losses at untreated snow could be helpful. What is the significance of the symbols (e.g. circles), do they just indicate the lines or are they measurement points?

5) The discussion of the assumed reaction system is unclear: Why should be only 0.5 ozone molecules consumed per bromine molecule (Br2)? Reaction equations 1 through 3 suggest that it is at least 2 ozone molecules. The disproportionation reaction (BrO2- + BrO- ?) is missing from the scheme. What is the meaning of 'assuming a net loss of 1 ozone molecule per bromide molecule'? And how is the number of 1E16 available bromide ions calculated?

6) The discussion of available bromide vs. observed ozone loss (page 9, lines 227 ff)

[Figure]

states that the latter is much smaller than the former. Actually one could say that the observed ozone loss is three orders of magnitude larger than the calculated bromide flux. But what is the conclusion from this calculation?

7) Table 1 gives the bromide content of the samples in ppbw, while in most of the remaining manuscript bromide is given in micro M. It would be helpful to include both numbers. Also, the SSA is given per gram, which is fine, but the total snow surface area would also be good to know (difficult to calculate since the snow density is not given).

8) The conclusion section basically states that there is experimental evidence that aged snow (subjected to a temperature gradient) may essentially not release volatile bromine. This is an interesting finding, but it appears difficult to draw quantitative conclusions from this result. The speculations about switching off other reaction pathways (page 13, lines 347 ff) do not appear to follow from the reported findings.

9) In fact it would be interesting to know how long it actually takes to remove the reactivity of doped snow towards ozone. From the data given here it only follows that the reactivity is large at age zero and essentially zero at age 12 days. It would be interesting to know how large the reactivity is after e.g. 1, 4, 8 days. Likewise it would be interesting whether bromine is actually released to the gas phase. This could be found out by determining the bromide contents of the snow after the experiment.

In summary, this is an interesting paper, but for the rather small amount of data it is way too long, and not many conclusions can be drawn yet. The presentation could be made more clear and easier to read (see above) and in a number of places the text could be considerably shortened.

---

## Referee Comment (RC2) · Anonymous Referee #2 · 22 Jun 2020

The authors examine the ozone reactivity of bromide-doped laboratory "snow" and the effect of temperature-gradient metamorphism. They find that metamorphism shuts down the reactivity, apparently because it buries the bromide away from the air-ice interface. The paper is interesting because of its connection between snow physics and chemistry and implications for the reactivity of natural snow.

***Major point

Overall, the manuscript is interesting and deserves to be (eventually) published. But the writing of the manuscript is a problem: it is often difficult to follow, non-linear, and sometimes rambling. It needs significant attention from the first author but also the senior authors.

[Figure]

Page 9 is one example. First, the entire page is one paragraph, as occurs on a number of pages. It would be much better to break the text into smaller paragraphs, each with a main theme. Second, the discussion circles around and around, repeating topics (e.g., the assumed diffusion coefficient) rather than linearly dealing with one topic and then moving on to the next. It makes it difficult to follow the arguments. The paper is already short, but could probably be shortened (and strengthened) by removing repetition, using a more logical flow, and removing extraneous ideas.

\*\*Other points

Line 26: The text states "tropospheric O3 reduction", but this is misleading since the global tropospheric O3 mixing ratio is increasing. Better wording would be "tropospheric O3 sink".

l. 71. What is "environmental snow" and how is it different from Arctic snow?

l. 81. How were the artificial drops produced? Paint sprayer?

l. 84. Samples were annealed at -5 C for 7 days. Why such a long annealing time? Why the focus on minimizing grain boundaries?

l. 88. This line also discusses 7 days of annealing. Is this in addition to the 7 days described on line 84, or is the same annealing description repeated twice? Or were samples were annealed in the reactor tubes for 7 days?

l. 127. UV illumination of N2/O2 mixtures can also make NOx in addition to O3. Was there any attempt to detect whether NOx was formed? Any evidence of NOx reactions, e.g., formation of nitrate in the O3-exposed snow?

Figure 1. What are the solid lines that connect the symbols? Continuous ozone measurements? Are the symbols then just the continuous result at specific times or is it something else? Make it clear in the caption that "undoped" means no added bromide for the grey line. How much variability was there in O3 loss rates for the undoped samples?

l. 162. It's not clear what is meant by "Taken the large variation in the ozone loss of samples exposed to metamorphism..."

l. 168. This paragraph is not easy to understand. Part of the problem is starting with the "result", i.e., comparison of O3 loss rates, before describing all of the steps that make this comparison meaningful.

l. 183. The sentence that starts "In summary, we conclude..." seems out of place. It is not a summary of the previous portion of the paragraph and it is a point that was made (or at least implied) in the earlier discussion of Figure 1.

l. 186. Here the authors attribute the background loss rate to "ozone self-reaction", but is there any evidence from previous studies (e.g., in solution) that there is an appreciable O3-O3 reaction? This proposed mechanism is too specific given the lack of evidence. Also, on the next page the authors attribute the background loss to impurities, not to ozone-ozone reactions. The impurities hypothesis seems more likely.

l. 189. This is excellent agreement, but it's not "perfect", since current loss rate is up to 2x higher than the past rate.

l. 205. I wouldn't expect that the snow made here is acidic. Are there pH measurements of the melted solution?

l. 220. This low coverage isn't necessarily evidence that bromide loss limits ozone depletion, since it's difficult to compare a bromide surface coverage with a gas-phase ozone concentration. A clearer controlling factor is instead the number of ozone molecules lost compared to the number of bromide ions initially on the ice.

l. 228. What's the reference for this diffusion coefficient?

l. 243. Solutes have been found at grain boundaries, but the sentence indicates that this location is different from "micropockets". My understanding of micropockets is that represent any liquid-like inclusion within the ice matrix, including at grain boundaries. If the authors want to use "micropockets" in a more specific way, they should define the

term.

l. 250. The use of parentheses to indicate a parallel sentence meaning is confusing. Better to have two separate, clear sentences.

l. 254. It's not clear why this possible (but hugely uncertain) number of micropockets is interesting. The number is highly sensitive to the radius, of which we only know an approximately upper bound. Furthermore, the estimate of the number of micropockets seems to have little bearing on whether micropockets can be "excluded" as a major location for bromide. The ozone data indicates that approximately 3/4 of the bromide is present internally in ice, whether in micropockets, at grain boundaries (if these are not part of micropockets) or as solid solution. How does a highly uncertain micropocket number estimate change this?

l. 257. The ozone depletion data argues strongly against all of bromide being present at the air-ice interface. So why the need for lines 257-258 to say this point again?

l. 269. What does it mean that the snow is "fully connected"? The snow that was not exposed to the temperature gradient looks as "fully connected" as the one that was exposed. Is it not?

l. 280. What does it mean that the particles "show edge structures" and why is this important? It seems that any solid-gas system will have edges.

Table 1. What is n for each of the four conditions?

l. 302. Early only one diffusion coefficient was presented, but now the issue is presented again, but with a range of values. This range should be presented in the initial discussion of diffusivities and either a best value, bounding value, or range should be presented in both locations.

l. 302. "10-12" needs to be formatted (superscripted "-12").

l. 303. 40 - 220 nm/s is a diffusion rate or speed, not a distance.

l. 304. What is the ice-growth diffusion mechanism? It is not clear how the comparison of rates supports this (unspecified) mechanism.

If the rate of bromide diffusion is 1 - 2 orders of magnitude faster than the rate of ice growth, one would expect uniform bromide distribution throughout the growing ice, meaning also some at the interface. But the experiments suggest no Br- at the interface in the snow that experienced the temperature gradient.

One interesting link to previous work: past modeling in solution (Finlayson-Pitts and Tobias) has shown that Br- is enriched at the air-solution interface. But the results in Fig. 1 for the T-gradient snow indicate that bromide is not present at the air-ice interface.

l. 309. Aren't the micropockets (by definition) already covered with ice? If so, this discussion of ice growth on brine drops seems to miss the point.

l. 328. "Burial of acidic trace gases with atmospheric relevance has previously been discussed for these volatile species (Huthwelker et al., 2006)." Rather than give this vague statement, it would be better to make a specific statement about the past results that are relevant to the current work.

l. 331. Why use "only" to qualify the HCl uptake?

l. 346. "As a consequence. . ." This is an interesting point.

l. 369. This last sentence is no longer referring to bromide? If not, it does not fit with the rest of the paragraph.
* * *

---

## Author Comment (AC1) · 2 Aug 2020

The manuscript reports on a small series of six experiments quantifying the ozone loss on bromide-doped artificial snow samples. The effect of subjecting the samples to temperature gradients for extended periods of time (days) is studied.

This is an interesting and important study possibly allowing conclusions on the avail- ability of bromide and the processes at the ice-air interface in (aged) snow. It is, there- fore, in principle relevant for understanding and modelling bromine release events ob- served in polar regions.

The manuscript contains important information relevant to the readers of ACP and should be published. However it contains a number of deficiencies and significant improvements are possible and should be made.

We thank the referee for recognizing the interesting aspects of this manuscript and for the positive judgement on the significance. We have substantially shortened and streamlined the manuscript and hope that the revised version convinces you.

1) Frequently release of volatile bromine is mentioned, however the experiments really determine loss of ozone, this fact should be stated more explicitly.

Indeed, we observe ozone loss and try to argue carefully that we assign this loss to the well-known reaction of ozone with bromide in the dark. The fact that we observe ozone loss is now mentioned prominently in the abstract, introduction, and in the conclusion again.

Abstract: "... the heterogeneous reactive loss of ozone in the dark at a concentration of 5-6e$^{12}$ molecule cm$^{-3}$ is investigated in artificial, shock-frozen snow samples doped with 6.2 µM sodium bromide …"

End of introduction: "The objective of this study is to investigate the heterogeneous reactivity of bromide oxidation by gas-phase ozone in the dark. To assess the surface concentration of bromide and its change during temperature gradient metamorphism the gas-phase ozone loss is monitored in this study. Bromide concentration in the doped snow samples (6.2 µM) is typical for snow on Arctic sea ice (Pratt et al., 2013)"

Conclusion: «We have presented an assessment of the effects of metamorphism on the loss of gas-phase ozone in bromide doped snow. Experiments were performed in the dark in snow doped with 6.2 µM sodium bromide»

2) The data given in some parts are incomplete and are given in different units, so reading the manuscript requires a pocket calculator. For instance on page 5, lines 125-131 the air flow through the samples is given in ml/minute, the ozone mixing ratio in ppb, while later (page 9) the number of ozone molecules per second is required. Although the manuscript mentions release of bromine 'in light and in the dark' one assumes that the experiments were performed in the dark, but this is not said in the manuscript. Volumes are sometimes given in ml, sometimes in cm3.

Thank you for pointing this out. We have replaced ml by cm$^3$ throughout the document. We still give the ozone mixing ratio (in ppb) but only in parenthesis next to the gas-phase concentration in

molecule cm$^{-3}$ for the convenience of the reader. We also mention the chemistry in the dark more explicitly. Thank you for this comment. We'd like to point out -however- that we use the well-known heterogeneous chemistry of bromide with ozone in the dark to probe the availability and reactivity of bromide at the snow-air interface. We are convinced that our results can also be directly be applied to the photochemistry of bromide with ozone.

Conclusion: "We have presented an assessment of the effects of metamorphism on the loss of gas-phase ozone in bromide doped snow. Experiments were performed in the dark in snow doped with 6.2 µM sodium bromide.»

Abstract: " For this, the heterogeneous reactive loss of ozone in the dark at a concentration of 5-6E12 molecule cm$^{-3}$ is investigated in artificial, shock-frozen snow samples doped with 6.2 µM sodium bromide and with varying metamorphism history."

End of introduction: "The objective of this study is to investigate the heterogeneous reactivity of bromide oxidation by gas-phase ozone in the dark. To assess the surface concentration of bromide and its change during temperature gradient metamorphism the gas-phase ozone loss is monitored in this study. Bromide concentration in the doped snow samples (6.2 µM) is typical for snow on Arctic sea ice (Pratt et al., 2013)"

3) A table is missing, which summarizes the pertinent data of the experiments: Volume of the reaction chamber, flow rate, snow density, snow surface area, number of ozone molecules lost per second, etc..

We have expanded the table. We prefer not to give give the number of ozone molecules lost per second as this number changes with time. For more data we kindly refer to the data repository where data tables can be downloaded.

**Table 1: Settings for the snow samples;** The number of days gives the duration of metamorphism. Br denotes the concentration of bromide as derived by ion chromatography; SSA is the specific surface area as derived from the microCT scans with an typical error of ± 6% (Kerbrat et al., 2008). The density was derived based on the weight of the snow sample and the volume of the sample holder. The mass denotes the amount of snow during the ozone exposure experiments and the surface area denotes the total surface area of the snow during the ozone exposure experiments. Each experiment with added bromide and an average of the 5 experiments with no added bromide is shown in Figure 1 and discussed in this work.

| | Br [µM] | SSA [cm g$^{-1}$] | density of snow [g cm$^{-3}$] | mass [g] | Surface area [cm$^2$] |
|---|---|---|---|---|---|
| 0 days | 6.2 | 183 | 0.33 | 17 | 3118 |
| 0 days | 6.2 | 183 | 0.32 | 11 | 2018 |
| 12 days, 31 °C cm$^{-1}$ gradient | 6.2 | 162 | 0.41 | 14 | 2268 |
| 12 days, isothermal | 6.2 | 143 | 0.45 | 16 | 2281 |
| 12 days, isothermal | 6.2 | 143 | 0.35 | 14 | 1996 |
| 0 days | <0.12 | 195 | 0.35 | 13 | 2540 |

| | | | | | |
|---|---|---|---|---|---|
| 0 days | <0.12 | 195 | 0.3 | 10 | 1953 |
| 0 days | <0.12 | 176 | 0.3 | 12 | 2113 |
| 12 days, 31 °C cm$^{-1}$ gradient | <0.12 | 167 | 0.371 | 14 | 2336 |
| 12 days, 31 °C cm$^{-1}$ gradient | <0.12 | 167 | 0.390 | 17 | 2836 |

4) Fig. 1: The figure summarizes all experimental findings of the manuscript, therefore it should be as informative and clear as possible. However, it is actually quite hard to read since most of the data are huddled in the lowest 20% or so of the plot. It would be helpful if the plot could be split in two, one ranging to 8E12 molec/s or even higher (what are actually the highest measured ozone loss rates?), one showing the data up to e.g. 3E12. Also additional lines indicating the ratio of losses at treated snow vs. losses at untreated snow could be helpful. What is the significance of the symbols (e.g. circles), do they just indicate the lines or are they measurement points?

Thank you for pointing out difficulties in reading the Figure. With all respect, we believe that the important parts of the figure are readable. Nevertheless, we have added a zoom to the figure. Thank you very much for this idea, it allowed us to also plot the standard deviation of the repeated experiments. We apologize for not being clear about the symbols. The ozone was monitored continuously, the symbols are just a help to differentiate the lines better. The figure is now introduced by:

"Figure 1 shows ozone loss rates for snow samples prior to and after exposure to dry metamorphism. The ozone loss rate was derived based on observed changes in gas-phase ozone concentration downstream of the flow tube packed with the snow sample. The ozone loss curves can be classified into three regions:

1. All samples show a high (> 9E12 molecule s$^{-1}$) loss rate during the initial period of ozone exposure up to 500 s.  This observed loss is attributed to the reaction of ozone with traces of impurities, to a delay by switching the gas flows, and to the residence time of the ozone gas in the porous snow and is not further analysed.
2. In the intermediate time regime, the ozone loss rate is largest for the two samples doped with 6.2 µM bromide prior to ageing under laboratory-controlled temperature gradient metamorphism with 4E12 molecule s$^{-1}$ and 7E12 molecule s$^{-1}$ at 1000 s duration of ozone exposure (Fig. 1, blue lines, open circles).  The loss rate was reduced by a factor of about 4 - 7 in the snow sample that experienced temperature gradient metamorphism with 1E12 molecule s$^{-1}$ at 1000 s duration of ozone exposure (Fig 1, yellow line, open square).....»
3. After about 8000 s ozone exposure, the ozone loss rates of all experiments approach zero loss of ozone. The raw data curves levelled off approaching a steady loss rate of 1.1-1.9E12 molecule s$^{-1}$. This background loss rate may be attributed to the reactive uptake of ozone  to ice driven by a self-reaction on the ice surface (Langenberg and Schurath, 1999), which is the main phase in the frozen solution samples investigated here. Langenberg and Schurath (1999) described a reactive ozone uptake coefficient on ice of 7.7-8.6E-9 at -15 °C and at ozone gas-phase concentrations similar to our work. The uptake coefficient normalizes the loss rate to the collision rate of ozone with the ice (or snow) surface. A loss rate of 0.86-0.90E12 molecule s$^{-1}$ can be derived based on the reported uptake coefficient for the experimental conditions of our doped samples prior to metamorphism, in good

agreement with our observations. Because this loss rate is not related to the bromide in the samples, it has been subtracted from the data discussed and shown in Fig. 1.

In Figure 1, the part of the data that is not used to quantify the ozone loss is now plotted faded and the y-axis scale is adopted to the fully show the range of the data used in the analysis

[Figure]

**Figure** 1: **Ozone loss rate with duration of exposure.** The snow samples with a bromide concentration of 6.2 µM experienced 0 days (blue lines, open circles) and 12 days (yellow line, open squares) of metamorphism with a temperature gradient of 31 °C m$^{-1}$. The lower panel is a zoom to the data. Ozone data were recorded continuously (lines), the markers are guides. The dotted lines are guide to the eyes, for periods where ozone loss data are not available (see text for details). Also shown are the ozone loss rates of snow samples after 12 days of isothermal metamorphism at -20 °C (red lines, open triangles). The grey line (open diamonds) denotes the average ozone loss rates of 5 samples with no bromide added and with and without exposure to temperature gradient metamorphism. The shaded area in the lower panel shows the standard deviation. The gas phase mixing ratio of ozone varied between 4.7-6.2 $10^{12}$ molecule cm$^{-3}$ for individual samples. Temperature during ozone exposure was -15 °C. At time 0, ozone in the carrier gas was passed over the snow samples.

5) The discussion of the assumed reaction system is unclear: Why should be only 0.5 ozone molecules consumed per bromine molecule (Br2)? Reaction equations 1 through 3 suggest that it is at least 2 ozone molecules. The disproportionation reaction (BrO2- + BrO- ?) is missing from the scheme. What is the meaning of 'assuming a net loss of 1 ozone molecule per bromide molecule'? And how is the number of 1E16 available bromide ions calculated?

Our apologies for not being clear. We have rewritten this section:

«Generally, the products and reaction mechanism of the bromide oxidation by ozone in the aqueous phase strongly depend on reaction time, reactant concentration and pH (Haag and Hoigne, 1983; Heeb et al., 2014). For non-acidified conditions, as in our study, hypobromous acid ($HOBr/OBr^-$) is the main product (Eq. 1) that may react further with ozone (Eq. 2) to form bromite ($BrO_2^-$), disproportionate to bromide ($Br^-$) and bromate ($BrO_3^-$), or self-react to dibromine monoxide ($Br_2O$) (Heeb et al., 2014). Despite uncertainties in the precise product distribution in this study, ozone is lost in our study in the initial reaction with bromide and to some extent in the subsequent oxidation of hypobromous acid to bromite resulting in 1-2 ozone molecules lost per bromide ion. In particular, at acidic conditions relevant for atmospheric waters and ices, but not applicable to our experimental settings (Abbatt et al., 2012; Bartels-Rausch et al., 2014), bromine is formed and released to the atmosphere in a sequence of reaction steps (Eqs. 1 and 3).

$Br^- + O_3 \longrightarrow OBr^- + O_2$ (Eq. 1)
$OBr^- + O_3 \longrightarrow BrO_2^- + O_2$ (Eq. 2)
$OBr^- + Br^- + H^+ \longrightarrow Br_2 + OH^-$ (Eq. 3)

Thus, assuming a net loss of 1 ozone molecule per bromide molecule, one might estimate about 0.9 and 1.7 molecules of bromide are available for the multiphase reaction with ozone in the two porous snow samples prior to metamorphism. Assuming a net loss of 2 ozone molecules, 1.8 and 3.3E16 molecules of available bromide can be estimated for the two samples.»

6) The discussion of available bromide vs. observed ozone loss (page 9, lines 227 ff) states that the latter is much smaller than the former. Actually one could say that the observed ozone loss is three orders of magnitude larger than the calculated bromide flux. But what is the conclusion from this calculation?

With all respect, we tried to say exactly what you state. The larger observed ozone loss means that the smaller amount of bromide available to react is too small to explain this ozone loss. We keep the original text as follows, but hope that due to the added introduction to the location of impurities in snow, the point is clearer now:

"The striking loss of heterogeneous reactivity during temperature gradient metamorphism raises the question of the location of the reactive bromide in the shock-frozen, artificial snow samples before metamorphism. Snow can host impurities in several compartments (Bartels-Rausch et al., 2014): Chemical species, besides water, and ions can molecularly embedded within the ice matrix (solid-solution), molecularly adsorbed at the air-ice interface, in liquid or solid patches at the air-ice interface, in micropockets within the ice matrix including the ice-ice interface (at grain boundaries). Clearly, only bromide in direct contact with the gas phase, that is located at the air-ice interface or within the bulk at a distance that allows sufficient diffusion to the interface, is accessible to gas-phase ozone and thus reactive. In the following, we elaborate on the feasibility of bromide being hosted in these distinct departments in the samples used here.»

"Shock freezing aqueous solutions may preserve the homogeneous distribution of solutes also in the ice matrix from where the bromide might diffuse to the air-ice interface and heterogeneously react with the ozone. One may estimate that the total amount of bromide diffusing from the ice bulk to the surface is 0.2 - 1.6E10 molecule per second. This is much less than the ozone loss observed in our experiments clearly showing that the bromide is not present homogeneously in the ice matrix of the snow samples.»

7) Table 1 gives the bromide content of the samples in ppbw, while in most of the remaining manuscript bromide is given in micro M. It would be helpful to include both numbers. Also, the SSA is given per gram, which is fine, but the total snow surface area would also be good to know (difficult to calculate since the snow density is not given).

Fixed.

8) The conclusion section basically states that there is experimental evidence that aged snow (subjected to a temperature gradient) may essentially not release volatile bromine. This is an interesting finding, but it appears difficult to draw quantitative con- clusions from this result. The speculations about switching off other reaction pathways (page 13, lines 347 ff) do not appear to follow from the reported findings.

Here, we kindly disagree and would like to point out that we have shown:

* ozone is lost from the gas-phase in samples that have been doped with bromide prior to temperature gradient metamorphism. The loss rate agrees well with previous studies. Giving, despite the small number of experiments, confidence in the loss being driven by reaction of ozone with bromide.

* to explain the observed ozone losses, bromide needs to be present at the air-ice interface in higher amounts than predicted for a homogeneous distribution in the ice. This agrees to earlier studies. As the exclusion of impurities during freezing is certainly a question of concentration, relevance comes from the fact that this is the first chemical study with uM of bromide that is the same concentration as found in the Arctic.

* the ozone reactivity decreases after temperature gradient metamorphism. In fact, we find no bromide driven ozone loss at all in the aged snow. Main reason being that diffusion is just too slow. The life time of bromide in snow assuming there is a sink at the surface, so the flux out of ice is limited by diffusion, is in the order of years. This is the main conclusion of general relevance. Quantifying the loss further is then rather a question of how reactive the snow was prior to metamorphism which is a question on the location and thus sources of the bromide in the snow.

9) In fact it would be interesting to know how long it actually takes to remove the reactivity of doped snow towards ozone. From the data given here it only follows that the reactivity is large at age zero and essentially zero at age 12 days. It would be inter- esting to know how large the reactivity is after e.g. 1, 4, 8 days. Likewise it would be interesting whether bromine is actually released to the gas phase. This could be found out by determining the bromide contents of the snow after the experiment.

We fully agree. This study directly links changes in chemical reactivity to snow metamorphism for the first time at low concentration that are not only relevant for Earth's Arctic environment, but also low enough to potentially allow the formation of a solid-solution. Certainly, investigating how many re- crystallization cycles - by varying the time or the temperature gradient - are a follow up that we also highly recommend.

In summary, this is an interesting paper, but for the rather small amount of data it is way too long, and not many conclusions can be drawn yet. The presentation could be made more clear and easier to read (see above) and in a number of places the text could be considerably shortened.

We have considerable shortened the manuscript by rearranging, removing repetitions, and removing some content. We like to point out, that discussing the phase diagram and the freezing point

depression is crucial when working with binary ice - salt mixtures and we'd argue that this discussion might indeed be longer than the presentation of the data itself. Discussing the phase diagram not carefully enough, has lead to false conclusions about the observed chemistry in the past (see Huthwelker , 2006).

This paper has only one conclusion: That heterogeneous reactivity is lost during snow metamorphism. We strongly argue that the total loss of heterogeneous chemistry after temperature gradient metamorphism is worth publishing. We think that this is one and clear conclusion from this work, not more but not less.

In a follow up study, we would suggest to use clean snow as for example prepared under nature identical conditions in the Anastasio group and dope it with bromide – either in aerosol deposits or molecularly adsorbed. Then a study with more parameter variation as you suggested and including changes to the temperature gradient should be done.

---

## Author Comment (AC2) · 2 Aug 2020

Reply to **Anonymous Referee #2**

The authors examine the ozone reactivity of bromide-doped laboratory "snow" and the effect of temperature-gradient metamorphism. They find that metamorphism shuts down the reactivity, apparently because it buries the bromide away from the air-ice interface. The paper is interesting because of its connection between snow physics and chemistry and implications for the reactivity of natural snow.

***Major point

Overall, the manuscript is interesting and deserves to be (eventually) published. But the writing of the manuscript is a problem: it is often difficult to follow, non-linear, and sometimes rambling. It needs significant attention from the first author but also the senior authors.

We thank the referee for detailed discussion of our manuscript and the generally positive feedback on the scientific quality and significance. We have significantly restructured and shortened the text removing parts that you might have felt are extraneous. We hope that you agree about the modified manuscript meeting the standards of ACP.

Page 9 is one example. First, the entire page is one paragraph, as occurs on a number of pages. It would be much better to break the text into smaller paragraphs, each with a main theme. Second, the discussion circles around and around, repeating topics (e.g., the assumed diffusion coefficient) rather than linearly dealing with one topic and then moving on to the next. It makes it difficult to follow the arguments. The paper is already short, but could probably be shortened (and strengthened) by removing repetition, using a more logical flow, and removing extraneous ideas.

Thanks for pointing to the long paragraphs. These have slipped our attention when uploading the manuscript. Shorter paragraphs and subheading have been introduced throughout the manuscript. The initial reason for discussing the diffusion at two different places of the manuscript was to clearly differentiate processes occurring during the ozone experiments and during the metamorphism as both differ in experimental settings such as temperature which impacts the diffusion. In the revised version, diffusion is no longer discussed in the context of the metamorphism and the discussion of diffusion is thus at one place:

"One may estimate that the total amount of bromide diffusing from the ice bulk to the surface is 0.2 - 1.6e10 molecules each second. This is much less than the ozone loss observed in our experiments clearly showing that the bromide is not present homogeneously in the ice matrix of the snow samples. Due to lack of diffusion coefficients of bromide in ice, the diffusion coefficients of $HNO_3$ in crystalline ice at -15 °C of 100e-12 cm2 s-1 (Thibert and Dominé, 1998) was used as upper limit and a diffusion coefficient of HCl at -15 °C of 3e-12 cm2 s-1 as lower bound was used in this calculation. Further, the aqueous concentration of 6.2 μM and the specific surface area of each snow sample as derived by the microCT data (Table 1) was used."

**Other points

Line 26: The text states "tropospheric O3 reduction", but this is misleading since the global tropospheric O3 mixing ratio is increasing. Better wording would be "tropospheric O3 sink".

Thank you! Fixed.

"Recent improvement in global atmospheric chemistry models indicate that halogen chemistry accounts for about 14% of the global tropospheric ozone sinks (Schmidt et al., 2016)"

l. 71. What is "environmental snow" and how is it different from Arctic snow? l. 81. How were the artificial drops produced? Paint sprayer?

We agree that his statement was confusing. We focus on Arctic snow in the revised version:

"Bromide concentration in the doped snow samples (6.2 µM) is typical for snow on Arctic sea ice (Pratt et al., 2013)."

l. 84. Samples were annealed at -5 C for 7 days. Why such a long annealing time? Why the focus on minimizing grain boundaries?
l. 88. This line also discusses 7 days of annealing. Is this in addition to the 7 days described on line 84, or is the same annealing description repeated twice? Or were samples were annealed in the reactor tubes for 7 days?

Thank you for pointing us to this inconsistency. The samples were stored once at -5°C. Grain boundaries have a pronounced impact on snow physics and physical chemistry. They may host impurities, that were expelled during freezing to the ice-ice interface and may act as shortcut for diffusion. To give the ice particles time to anneal leading to a reduction in the grain boundaries, we stored the samples at -5°C for about a week.

"The samples were left overnight at –45°C and then, stored isothermally at –5 °C for 7 days to anneal and to minimize grain-boundaries (Blackford, 2007; Riche et al., 2012). The samples were returned to - 45 °C after this isothermal treatment to slow down further changes with time and stored up to 54 days at - 45 °C prior to the metamorphism experiments for logistic reasons."

l. 127. UV illumination of N2/O2 mixtures can also make NOx in addition to O3. Was there any attempt to detect whether NOx was formed? Any evidence of NOx reactions, e.g., formation of nitrate in the O3-exposed snow?

We have focused on the ozone loss as observable and have not tried to detect NOx or nitrates. We would like to note that the O2 was illuminated at 185 hv – which photolyzes O2 but not N2. Since we routinely use O3 to titrate NO for NO2 calibration, we note that our UV O3 generator is NOx free.

Figure 1. What are the solid lines that connect the symbols? Continuous ozone measurements? Are the symbols then just the continuous result at specific times or is it something else? Make it clear in the caption that "undoped" means no added bromide for the grey line. How much variability was there in O3 loss rates for the undoped samples?

The ozone was recorded continuously, and the symbols were only added to help differentiate the lines (in black and white prints). We have modified the figure and caption to make this clearer, also adding the standard deviation to show the variability in the O3 loss of undoped snow.

[Figure]

**Figure 1: Ozone loss rate with duration of exposure**. The snow samples with a bromide concentration of 6.2 µM experienced 0 days (blue lines, open circles) and 12 days (yellow line, open squares) of metamorphism with a temperature gradient of 31 °C m$^{-1}$. The lower panel is a zoom to the data. Ozone data were recorded continuously (lines), the markers are guides. The dotted lines are guide to the eyes, for periods where ozone loss data are not available (see text for details). Also shown are the ozone loss rates of snow samples after 12 days of isothermal metamorphism at -20 °C (red lines, open triangles). The grey line (open diamonds) denotes the average ozone loss rates of 5 samples with no bromide added and with and without exposure to temperature gradient metamorphism. The shaded area in the lower panel shows the standard deviation. The gas phase mixing ratio of ozone varied between 4.7-6.2e12 molecules cm-3 for individual samples. Temperature during ozone exposure was -15 °C. At time 0, ozone in the carrier gas was passed over the snow samples.

l. 162. It's not clear what is meant by "Taken the large variation in the ozone loss of samples exposed to metamorphism. . ."

We agree and have removed this statement.

l. 168. This paragraph is not easy to understand. Part of the problem is starting with the "result", i.e., comparison of O3 loss rates, before describing all of the steps that make this comparison meaningful.

We have reworded the beginning of the paragraph and hope it is easier to follow.

"The reaction of gas-phase ozone with frozen solutions containing bromide has been studied in great detail previously (Wren et al., 2010; Oldridge and Abbatt, 2011; Abbatt et al., 2012; Wren et al., 2013). Oldridge and Abbatt (2011) described coated wall flow tube studies on frozen sodium bromide/sodium chloride/water mixtures at -15°C and Wren et al. (2010) reported on a laser-induced fluorescence study with sodium bromide/water mixtures at

- 20°C. The studies by Wren et al. (2010) and by Oldridge and Abbatt (2011) were done with an initial sodium bromide concentration of 10 mM and a gas-phase ozone concentration of $1\times10^{14}$ molecule cm$^{-3}$ and $80\times10^{14}$ molecule cm$^{-3}$, respectively. Oldridge and Abbatt (2011) have argued that this multiphase reaction proceeds in the liquid fraction of sample containing bromide-brine that is in equilibrium with ice between 0 °C and the eutectic temperature where the salt precipitates. The eutectic temperature of sodium bromide is at or below -28 °C (Stephen and Stephen, 1963). ... Despite the differences in the concentration of bromide in the solutions used to freeze the films, the similar concentration of bromide in the brine during ozone exposure makes a comparison of the experimental results feasible. For the comparison, the reported uptake coefficients of $1.5\times10^{-8}$ and 4-$2\times10^{-8}$, respectively (Wren et al., 2010; Oldridge and Abbatt, 2011), were transferred...»

l. 183. The sentence that starts "In summary, we conclude. . ." seems out of place. It is not a summary of the previous portion of the paragraph and it is a point that was made (or at least implied) in the earlier discussion of Figure 1.

Yes, we agree. Thank you. We have removed the sentence and restructured the paragraph.

l. 186. Here the authors attribute the background loss rate to "ozone self-reaction", but is there any evidence from previous studies (e.g., in solution) that there is an appreciable O3-O3 reaction? This proposed mechanism is too specific given the lack of evidence. Also, on the next page the authors attribute the background loss to impurities, not to ozone-ozone reactions. The impurities hypothesis seems more likely.
l. 189. This is excellent agreement, but it's not "perfect", since current loss rate is up to 2x higher than the past rate.

We apologize for not being clear. We have rewritten this paragraph where we discuss the reactive loss of ozone on pure ice. As ice is the main surface in our samples, we conclude that this "ozone self-reaction" on ice may explain the long-lasting tail of the data. Of course, impurities might contribute, however, we think that these react away faster. We also describe the agreement as good and not perfect in the revised version.

« 3.	After about 8000 s ozone exposure, the ozone loss rates of all experiments approach zero loss of ozone. The raw data curves levelled off approaching a steady loss rate of 1.1-1.9e12 molecule s-1. This background loss rate may be attributed to the reactive uptake of ozone to ice driven by a self-reaction on the ice surface (Langenberg and Schurath, 1999), which is the main phase in the frozen solution samples investigated here. Langenberg and Schurath (1999) described a reactive ozone uptake coefficient on ice of 7.7-8.6e-9 at -15 °C and at ozone gas-phase concentrations similar to our work. The uptake coefficient normalizes the loss rate to the collision rate of ozone with the ice (or snow) surface. A loss rate of 0.86-0.90e12 molecules s-1 can be derived based on the reported uptake coefficient for the experimental conditions of our doped samples prior to metamorphism, in good agreement with our observations. Because this loss rate is not related to the bromide in the samples, it has been subtracted from the data discussed and shown in Fig. 1.»

l. 205. I wouldn't expect that the snow made here is acidic. Are there pH measure- ments of the melted solution?

We agree. We do not expect the snow to be acidic. We have not measured the pH of the molten snow as this would say little about the pH of the reactive medium.

l. 220. This low coverage isn't necessarily evidence that bromide loss limits ozone depletion, since it's difficult to compare a bromide surface coverage with a gas-phase ozone concentration. A clearer controlling factor is instead the number of ozone molecules lost compared to the number of bromide ions initially on the ice.

Thank you, we agree and have removed this section.

We always referred to Dominé's work. This section was rewritten as mentioned above.

l. 243. Solutes have been found at grain boundaries, but the sentence indicates that this location is different from "micropockets". My understanding of micropockets is that represent any liquid-like inclusion within the ice matrix, including at grain boundaries. If the authors want to use "micropockets" in a more specific way, they should define the term.

We agree fully with this definition and have reworded to make this clearer.

"Chemical species, besides water, and ions can molecularly embedded within the ice matrix (solid-solution), molecularly adsorbed at the air-ice interface, in liquid or solid patches at the air-ice interface, in micropockets within the ice matrix including the ice-ice interface (at grain boundaries).»

We have also removed the detailed discussion on micropockets and now simply state:

"We propose that the brine forms liquid patches on the surface and filaments along the grain boundaries at the interface as observed for higher concentrated frozen salt solutions (Blackford et al., 2007). A homogenous film covering the total snow surface is unlikely: A back-of-the-envelope calculation with the total amount of bromide doped to the samples and with a concentration of 3.4 M gives a brine layer with a thickness of only ~0.1 nm at -15°C for the specific surface area of the doped snow samples. This is unfeasible, because the thickness of an ice monolayer is roughly 0.3 nm. Whether the unreactive fraction of the bromide is located in a solid solution or in micropockets within the ice matrix is beyond the scope of this work, both compartments explain its non-reactivity."

l. 250. The use of parentheses to indicate a parallel sentence meaning is confusing. Better to have two separate, clear sentences.

Thank you, we have removed the discussion of diffusion during metamorphism.

"One may estimate that the total amount of bromide diffusing from the ice bulk to the surface is 0.2 - 1.6e10 molecules per second. This is much less than the ozone loss observed in our experiments clearly showing that the bromide is not present homogeneously in the ice matrix of the snow samples. Due to lack of diffusion rates of bromide in ice, the diffusion rates of $HNO_3$ in crystalline ice at -15 °C of 100 $e^{-12}$ $cm^2$ $s^{-1}$ (Thibert and Dominé, 1998) was used as upper limit and a diffusion rate of HCl at -15 °C of 3$e^{12}$ $cm^2$ $s^{-1}$ as lower bond was used in this calculation. Further, the aqueous

concentration of 6.2 µM and the specific surface area of each snow sample as derived by the microCT data (Table 1) was used."

l. 254. It's not clear why this possible (but hugely uncertain) number of micropockets is interesting. The number is highly sensitive to the radius, of which we only know an approximately upper bound. Furthermore, the estimate of the number of micropockets seems to have little bearing on whether micropockets can be "excluded" as a major location for bromide. The ozone data indicates that approximately 3/4 of the bromide is present internally in ice, whether in micropockets, at grain boundaries (if these are not part of micropockets) or as solid solution. How does a highly uncertain micropocket number estimate change this?

We have removed this section to make the manuscript more linear and clearer.

l. 257. The ozone depletion data argues strongly against all of bromide being present at the air-ice interface. So why the need for lines 257-258 to say this point again?

With all respect, we think that the question whether or not impurities form a homogeneous film on snow or ice is still raising intensive discussions and is worth mentioning. (F. Dominé et al., J. Phys. Chem. A, **117**, 4733-4749 (2013).) We have shortened the manuscript elsewhere and hope that this supporting argument now won't disturb the reading.

l. 269. What does it mean that the snow is "fully connected"? The snow that was not exposed to the temperature gradient looks as "fully connected" as the one that was exposed. Is it not?

Prior to temperature gradient the individual spheres that have been originally frozen still dominate the picture. The referee is right, that there are connections between the individual particles from the start due to fast sintering. But these bonds are weak and limited to a few points of contact. Fully connected referred to the individual spheres no longer being visible in the snow structure. We have reworded the section to make it clearer:

"In the microCT image of the snow sample prior to metamorphism individual spheres with 300 – 600 µm diameter are visible (Fig. 2, upper image). With developing snow metamorphism, the spheres get increasingly bonded and a new porous snow structure forms, while the recognition of the individual snow particles is lost (Fig. 2, lower image)."

l. 280. What does it mean that the particles "show edge structures" and why is this important? It seems that any solid-gas system will have edges.

We thank the referee for this comment. The appearance of edged structures is typical for temperature gradient metamorphism and we thought that it has occurred to some extent during isothermal storage or transport of the sample.

After reconsideration and comment of one of our co-authors, we agree with the referee. The angular structures may also have formed during isothermal storage. The minimal energy surface represents a polyhedron with rounded corners. (Löwe, H., Spiegel, J., & Schneebeli, M. (2011). Interfacial and structural relaxations of snow under isothermal conditions. Journal of Glaciology, 57(203), 499-510. doi:10.3189/002214311796905569). Thus, the faceted structures visible in Fig. 2a) are indeed formed during isothermal metamorphism.

We have deleted the particular section in the manuscript.

Table 1. What is n for each of the four conditions?

Table 1 now lists all individual experiments and we hope it is clearer that n is 1 for all but the 5 experiments without added bromide that were averaged in Figure 1.

l. 302. Early only one diffusion coefficient was presented, but now the issue is presented again, but with a range of values. This range should be presented in the initial discussion of diffusivities and either a best value, bounding value, or range should be presented in both locations.
l. 302. "10-12" needs to be formatted (superscripted "-12").
l. 303. 40 - 220 nm/s is a diffusion rate or speed, not a distance.

Thank you, all points fixed.

l. 304. What is the ice-growth diffusion mechanism? It is not clear how the comparison of rates supports this (unspecified) mechanism.
If the rate of bromide diffusion is 1 - 2 orders of magnitude faster than the rate of ice growth, one would expect uniform bromide distribution throughout the growing ice, meaning also some at the interface. But the experiments suggest no Br- at the interface in the snow that experienced the temperature gradient.
One interesting link to previous work: past modeling in solution (Finlayson-Pitts and Tobias) has shown that Br- is enriched at the air-solution interface. But the results in Fig. 1 for the T-gradient snow indicate that bromide is not present at the air-ice interface.

The ice-growth diffusion mechanism has been discussed in our previous work. Basically, it addresses the question, why the reactivity decreases while the water of the snow undergoes 5 complete recrystallisation cycles. During one full cycle, any bromide patches initially covered by the growing ice might be exposed to the air-ice interface again, when the water that covered them evaporates towards the end of the complete recrystallisation cycle. The detailed mechanism is speculative, so we removed it from the discussion of this manuscript and state the result as the apparent ability of the temperature gradient metamorphism to foster re-distribution of impurities into the interior of the ice matrix.

When a homogeneous distribution of bromide is reached, some of it would end up at the interface and at a distance from where it could diffuse to the interface. We discuss this aspect now early on in the manuscript where we conclude that the amount of bromide in this region is too small to give noticeable ozone consumption:

"Shock freezing aqueous solutions may preserve the homogeneous distribution of solutes also in the ice matrix from where the bromide might diffuse to the air-ice interface and heterogeneously react with the ozone. In the following, this reacto-diffusive loss is estimated. Due to lack of knowledge of the diffusion coefficient of bromide in ice, the diffusion coefficient of $HNO_3$ in crystalline ice at -15 °C of 100e-12 cm2 s-1 (Thibert and Dominé, 1998) was used as upper limit and a diffusion coefficient of HCl at -15 °C of 3e-12 cm2 s-1 as lower bound was used in this calculation. Further, the aqueous concentration of 6.2 µM and the specific surface area of each snow sample as derived by the microCT data (Table 1) was used. Based on these assumptions, one may estimate that the total amount of bromide diffusing from the ice bulk to the surface is 0.2 - 1.6e10 molecules each second. This is much less than the ozone loss observed in our experiments, clearly showing that the bromide is not present homogeneously in the ice matrix of the snow samples after shock freezing"

We prefer to not discuss the surface propensity of bromide in aqueous solution, which has been revisited recently (Gladich, I., Chen, S., Vazdar, M., Boucly, A., Yang, H., Ammann, M., and Artiglia, L.: Surface Propensity of Aqueous Atmospheric Bromine at the Liquid–Gas Interface, The Journal of Physical Chemistry Letters, 3422-3429, 10.1021/acs.jpclett.0c00633, 2020.

Olivieri, G., Parry, K. M., D'Auria, R., Tobias, D. J., and Brown, M. A.: Specific Anion Effects on Na+ Adsorption at the Aqueous Solution–Air Interface: MD Simulations, SESSA Calculations, and Photoelectron Spectroscopy Experiments, The Journal of Physical Chemistry B, 122, 910-918, 10.1021/acs.jpcb.7b06981, 2018.)

l. 309. Aren't the micropockets (by definition) already covered with ice? If so, this discussion of ice growth on brine drops seems to miss the point.

Thank you for pointing to this inconsistency. We have reworded as follows:

"Patches at the interface may also be covered by the growing ice in line with Nagashima et al. (2018), who observed preferential growth of ice onto brine droplets compared to the neat ice surface.»

l. 328. "Burial of acidic trace gases with atmospheric relevance has previously been discussed for these volatile species (Huthwelker et al., 2006)." Rather than give this vague statement, it would be better to make a specific statement about the past results that are relevant to the current work.

Thank you. We have rewritten the section to be clearer:

«Our observation of the ozone consumption showed that the bromide-doped snow samples lost their chemical reactivity towards gas-phase ozone during 12-days of temperature gradient metamorphism. This loss occurred without photochemistry forming volatile products. Post-depositional changes to bromide in snow have been observed in the field and have been explained by vivid photochemical reaction into volatile bromine. Volatile bromine might then be re-deposited on the snow surface after formation of more oxidized species, such as HOBr (Jacobi et al., 2002; Toom-Sauntry and Barrie, 2002). The burial of volatile trace gases into growing ice has also been discussed for acidic trace gases with atmospheric  relevance (Huthwelker et al., 2006).»

l. 331. Why use "only" to qualify the HCl uptake?

Thank you, fixed.
l. 346. "As a consequence. . ." This is an interesting point.

Thank you for this judgement.

l. 369. This last sentence is no longer referring to bromide? If not, it does not fit with the rest of the paragraph.

Indeed, the sentence does not refer to bromide alone:

"Based on our finding, another explanation would be a constant flux of bromide from the atmosphere refurbishing the bromide that is buried by temperature gradient metamorphism and thus providing reactive bromide at the air-ice interface.

This finding has significant environmental implications as it does not only stress the importance of the location of chemical species on their reactivity but shows that this location is rapidly changing in surface snow. One should note that incorporation of solutes into the interior of ice and snow makes them not only resistant to multiphase chemistry, but further reduces their tendency to be washed away by melt- or rain- water percolating the snow. The enrichment in the snow may thus contribute to later release of toxins to the marine food web upon the complete melting of the snow (Wania et al., 1998; Eichler et al., 2001; Steffen et al., 2008; Durnford and Dastoor, 2011; Grannas et al., 2013). Further, even under current warming conditions the buried species might be promising candidate for reconstructing past atmospheric composition from ice core records that have experienced melt effects (Eichler et al., 2001)."